# Characteristics of the Molecular Weight of Polyhexamethylene Guanidine (PHMG) Used as a Household Humidifier Disinfectant

**DOI:** 10.3390/molecules26154490

**Published:** 2021-07-26

**Authors:** Dong-Uk Park, Kee Won Yang, Jiwon Kim, Ju-Hyun Park, So-Yeon Lee, Kyung Ehi Zoh, Jung-Hwan Kwon, Soyoung Park, Han Bin Oh

**Affiliations:** 1Department of Environmental Health, Korea National Open University, Seoul 03087, Korea; jiwonk1012@mail.knou.ac.kr; 2Department of Chemistry, Sogang University, Seoul 04107, Korea; scr1109@gmail.com; 3Department of Statistics, Dongguk University, Seoul 04620, Korea; juhyunp@gmail.com; 4Humidifier Disinfectant Health Center, Department of Pediatrics, Asan Medical Center, College of Medicine, University of Ulsan, Seoul 05505, Korea; imipenem@hanmail.net; 5Department of Environmental Health Sciences, Graduate School of Public Health, Seoul National University, Seoul 08826, Korea; kezoh@snu.ac.kr; 6Division of Environmental Science and Ecological Engineering, Korea University, Seoul 02841, Korea; junghwankwon@korea.ac.kr; 7Department of Occupational and Environmental Medicine, Kangbuk Samsung Hospital, School of Medicine, Sungkyunkwan University, Seoul 03181, Korea; syoem.park@samsung.com

**Keywords:** humidifier-disinfectant-associated lung injury (HDLI), polyhexamethylene guanidine (PHMG), oligomer, molecular mass, polymerization, number-average molecular weight (M_n_), weight-average molecular weight (M_w_)

## Abstract

(1) Background: Household humidifier disinfectant (HD) brands containing polyhexamethylene guanidine (PHMG) have been found to cause the most HD-associated lung injuries (HDLIs) in the Republic of Korea. Nevertheless, no study has attempted to characterize the potential association of the health effects, including HDLI, with the physicochemical properties of PHMG dissolved in different HD brands. This study aimed to characterize the molecular weight (MW) distribution, the number-average molecular weight (M_n_), the weight-average molecular weight (M_w_), and the structural types of PHMG used in HD products. (2) Methods: Quantitative measurements were made using matrix-assisted laser desorption/ionization–time-of-flight mass spectrometry (MALDI-TOF MS). The M_n_, M_w,_ and MW distributions were compared among various HD products. (3) Results: The mean M_n_ and M_w_ were 542.4 g/mol (range: 403.0–692.2 g/mol) and 560.7 g/mol (range: 424.0–714.70 g/mol), respectively. The degree of PHMG oligomerization ranged from 3 to 7. The MW distribution of PHMG indicated oligomeric compounds regardless of the HD brands. (4) Conclusions: Based on the molecular weight distribution, the average molecular weight of PHMG, and the degree of polymerization, the PHMG collected from HDLI victims could be regarded as an oligomer. PHMG, as used in household humidifiers, should not be exempted from toxic chemical registration as a polymer. Further study is necessary to examine the association of PHMG oligomeric compounds and respiratory health effects, including HDLI.

## 1. Introduction

Many people who use household humidifiers containing chemical-based disinfectants (HDs) in the Republic of Korea (Korea) have developed lung injury, including chemical-associated asthma, interstitial pneumonitis, and widespread lung fibrosis [1,2,3,4]. These casualties are confirmed to be associated with the use of HDs. From 1994 to 2011, polyhexamethylene guanidine (PHMG), oligo (2-(2-ethoxy) ethoxyethyl guanidinium (PGH), and a mixture of chloromethylisothiazolinone (CMIT) and methylisothiazolinone (MIT) were widely used as HDs in Korea. The resulting lung injuries have been collectively named humidifier-disinfectant-associated lung injury (HDLI) [1,2,3]. 

As of April 2020, it has been officially verified that PHMG-containing HD products are associated with the most HDLI patients [5]. The container of HD with PHMG is labeled with instructions directing the user to add 10 mL of the HD to 2–3 L of water in a humidifier water tank. No information on the level of PHMG or the frequency of addition is included. Many people who use less HD than the amount instructed are reported to develop HDLI [5]. Other HD products containing PGH or a mixture of CMIT and MIT have the same background. To the best of our knowledge, there have been no reports of respiratory health problems, including lung injury, caused by chemicals containing PHMG and a mixture of CMIT and MIT used in industrial or consumer products as an anti-microbial additive.

Nevertheless, the physicochemical properties of the PHMG used in these HD products have been poorly defined and only rarely investigated. Specifically, PHMG is a family of polymers containing a guanidine subunit with high biocidal capacity against a range of microorganisms, while showing low toxicity to humans [6,7,8]. PHMG is synthesized by the condensation reaction of hexamethylenediamine (HMDA) and guanidine hydrochloride. Thus, the resulting PHMG polymeric mixture consists of molecules with different chain lengths and different end groups.

In a recent publication, we reported the properties of PHMG, including the concentrations and average molecular weights (MWs) of PHMG dissolved in HD product brands using matrix-assisted laser desorption/ionization–time-of-flight mass spectrometry (MALDI-TOF MS). The average MWs of PHMG in the HD products reported were in the range of 422–678 g/mol, indicating that the PHMGs used in HD brands are oligomeric compounds chaining less than eight monomer units [9]. 

PHMG consists of an isomeric polymeric guanidine group harboring an amino group, which contributes to its activity as a cationic antimicrobial macromolecule [10]. In addition, it is a water-soluble, odorless, colorless, and non-corrosive polymer and is less toxic to humans than other disinfectants currently in use [11]. It is known that each isomeric polymer that has a different end group has a different biocidal effect. Thus, it is considered worthwhile to examine the detailed isomeric molecular structures of PHMG and their contents (or components), which may be associated with biocidal activity and health effects. In this study, the MW distribution and isomeric structures of PHMG applied as an HD were further updated and characterized. The results will be helpful in better understanding the health effects caused by the use of PHMG-containing HD products.

## 2. Results

The mean M_n_ and M_w_ were 542.4 g/mol (range: 403.0–692.2 g/mol) and 560.7 g/mol (range: 424.0–714.70 g/mol), respectively (Table 1). The degree of PHMG oligomerization ranged from 3 to 7. The average MW and the degree of polymerization were not significantly different from one HD product to another (*p* > 0.05). In this study, the distribution of the PHMG MW is presented in intervals of 100 Da (Table 2). For eight HD brands, approx. 90% of PHMG was found to be distributed in the region lower than 800 g/mol. Figure 1 shows the structures of four different PHMG oligomeric types identified in our MALDI-TOF mass spectra. Types A, B, and E have one guanidine group in each unit, while type C has two. It is notable that type E has a cyclic structure in which two end groups are connected. Their relative contents were determined based on the abundances of the peaks in the MALDI-TOF mass spectra. The MW distribution was further categorized based on oligomer isomeric types A, B, C, and E, which have different end groups (Figure 2). The analysis of the MALDI-TOF mass spectra for all HD products revealed that the molecular structure types A and C are dominant, although there are some differences among HD brands. In general, differences in the distributions of isomer molecular types arise based on the reactants’ mix ratio between 1,6-hexamethylenediamine and guanidine hydrochloride and on reaction conditions, such as reaction temperature and time. In this study, only types A, B, C, and E were observed in the HD products. This observation is likely to be a result of the specific PHMG manufacturing conditions set by the manufacturers.

## 3. Discussion

PHMG used as an HD comprises oligomeric compounds chaining up to seven monomer units. Approximately 90% of PHMGs were distributed below 800 g/mol, although the MW distributions vary to some degree among HD brands (Table 2), indicating oligomeric compounds. Our MW distribution results are inconsistent with those found in the original document submitted to the National Institute of Environmental Research of Korea (NIER) by the manufacturer of the original PHMG raw material, SKYBIO 1125. It was reported to contain 25% PHMG for use in commercial HD products [12]. According to a document released in 1997 by NIER, SKYBIO 1125 has an M_n_ of 1274 g/mol, but no information on the MW distribution was provided. These data were based on measured values using gel-permeation chromatography (GPC). The discrepancy in M_n_ between our measurements for the HD products and the reported values may be due to the use of two different measurement methods: MALDI-TOF MS versus GPC. It has been reported that GPC-based measurements may overestimate the average molecular weights of oligomers or polymers [13]. PHMG was exempted from registration as a toxic chemical based on the submitted documents claiming that 27.4% (*w/w*) of PHMGs have molecular weights lower than 1000 g/mol and the products have hexamethylene diamine (HMDA) contents lower than 1% (*w/w*). PHMG was registered as a polymer macromolecule and an existing chemical without any evaluation of its inhalation toxicity under the Toxic Substances Control Law (TSCL) of Korea (enacted in 1991). PHMG was allowed for use as a carpet disinfectant at that time. 

In Australia, the PHMG contained in SKYBIO 1125 was also registered as a polymer, for which the M_n_ and M_w_ were reported to be 18,500 and 137,000 g/mol, respectively [14]. These values are far higher than the average MWs submitted to NIER (M_n_ of 1274 g/mol) and found in our study (M_n_ of 542.4 g/mol and M_w_ of 560.7 g/mol (Table 1 and Table 2). This PHMG product was registered as a microbial additive for plastics, fabric softeners, paints, swimming pools, paper, and sanitation in food processing plants and cooling towers. What could have caused such a massive discrepancy in the PHMG MWs between the submitted data and the experimental results is unclear.

SKYBIO 1125 is a raw ingredient in various commercial brands, including HD brands introduced in early 2000 and widely used until all HD products were removed from the market in 2011 due to the occurrence of an HDLI cluster. No official document has been identified that indicates how SKYBIO 1125, initially designed for industrial biocide products, became widely used for household HDs in Korea. A total of eight HD brands containing PHMG were sold in Korea between 2000 and 2011 [15]. Oxysaksak was the first marketed HD brand and accounted for the largest number (176, 39%) of HDLI casualties (*n* = 453) by a single brand [5]. As of the third round of the government investigation at the end of 2015, PHMG-containing HD products were the most frequently used HD products by HDLI patients (*n* = 234, 52%), followed by PGH HD products (*n* = 27, 6%) and products with a mixture of CMIT and MIT (*n* = 26, 6%) [5].

In PHMG-containing HD products, several components may be potentially related to health: PHMG oligomers, HMDA remaining unreacted as a raw reactant material, and several other additives. PHMG materials are composed of molecules with different chain lengths and different isomeric molecular structures with different end groups [16]. For example, in the HD products examined in this study, oligomers mainly range from 3 to 7 mer, and A and C types are the main structural types, while B and E types exist in low amounts. The complex physicochemical properties of PHMG, such as the molecular mass, degree of polymerization, and concentration of HMDA, may all be related to respiratory health effects, including asthma, lung injury, and more. The airborne level and MW distribution of PHMG could be assumed to be associated with health problems, including HDLI. Most of the inhaled PHMG dose can easily reach the respiratory system’s alveolar region, the target organ injured by HDs [17], due to the nanosize of HD aerosols dispersed in the air. The Korea Center for Disease Control reported that the average size of HD aerosols dispersed into the air ranges from 30 to 80 nm in a chamber experiment using an ultrasonic humidifier [18,19]. 

High concentrations of PHMG oligomer materials and their MW characteristics are likely to be two major factors causing severe health problems. We already reported the concentrations and MW distributions of PHMG among HD products we collected [9]. Interestingly, the MWs found in various HD brands were similar, despite the differences in concentrations between and within HD brands. The concentration range of PHMG dissolved in HD products ranged from 160 to 37,200 ppm and averaged 3100 ppm [20], which is far higher than the 13 μg/mL (ppm) suggested for a practical bactericidal effect, even when a diluted concentration is considered [21]. Wei et al. reported that an aqueous solution of PHMG with MW > 640 at a concentration as low as 1.0 ppm exhibits an antibacterial rate above 90%. The relationship between antibacterial activities and the MW of PHMG has been reported. Wei et al. reported all aqueous PHMG (M_w_ ≥ 516) solutions have excellent antibacterial activities (99.0%) against *S. aureus* and *P. aeruginosa* within 15 min when the concentration is increased to 5.0 ppm. This is different from polyhexamethylene biguanide (PHMB), another guanidine-containing polymer, which shows excellent antibacterial activity only when the polymerization degree is above 10. 

PHMG has the most antibacterial activity when the total positive charges of a single PHMG molecule match the negative charges of bacteria [16]. Oligoguanidine PHMG has excellent antibacterial activity [16]. The prevailing model for PHMG activity holds that guanidine kills bacteria through bacterial membrane damage, and the polymer does not interact with mammalian cell membranes. A polymer of low concern (PLC) is defined as a polymer that is deemed to have insignificant environmental and human health impacts, reducing regulatory requirements. However, some polymers cannot be regarded as PLCs when they contain certain elements or if they are water soluble, cationic, degradable, or hazardous, according to guidelines in many countries [22], regardless of the criteria for MW distribution of the polymers. In this sense, PHMG, which is water soluble and cationic, cannot be regarded as a PLC. In addition, the criteria for a polymer to be exempted from registration as a toxic chemical include the average MW and the level of monomer regardless of the type of polymer. Many countries implement this as a legal criterion to establish a polymer as a PLC. 

PHMG used in HDs is not eligible to be a PLC because it has water-soluble and cationic properties with net positive charges. Furthermore, this study found that the average percentage of PHMG oligomers with an MW of <500 (Table 1 and Table 2) is far higher than the composition criteria stipulated in Korea; polymers with M_n_ between 1000 and 10,000 must contain less than 10% oligomer contents of an MW below 500 g/mol and less than 25% oligomer content of an MW below 1000 g/mol [23]. The weight percentage of oligomers with an MW of <500 g/mol must be less than 5% in the United States and China and 2% in Japan and South Korea [24]. Based on the results found here, PHMGs used in HD brands are not PLCs and therefore should never have been registered as regular substances. To the best of our knowledge, no health effects have been caused by PHMGs used in products other than HDs. In addition, no study has been undertaken to examine the potential association of HDLI with PHMG MW properties, such as the average MW, MW distribution, and polymerization. Further research is necessary to examine the health effects of oligomeric PHMG and the relationship between the toxicity of PHMG and the molecular properties such as MW, the structural type, and salt type (phosphate or hydrochloride) [25,26,27] of PHMG.

A major limitation of this study was the considerable imbalance among the numbers of samples of the HD brands (Table 1 and Table 2, and Figure 2). We collected most of the HD samples through a field investigation of people who registered as patients with a national compensation program. The sample numbers of HD brands differ since the registered patients vary greatly in terms of HD brand used. In addition, the market volumes of different HD brands have been reported to vary greatly. However, there is no limitation on conclusions regarding the physicochemical properties of PHMG used in HDs, including the oligomeric compounds found in this study.

## 4. Materials and Methods

### 4.1. Collection of Humidifier Disinfectant (HD) Samples

The methods for collecting HD brands and their samples have been described elsewhere [28,29]. Most HD samples were collected from people who registered with the HDLI Investigation and Decision Committee (HLIIDC) program. Samples were stored in PE bottles, transported in an icebox, and stored in a refrigerator (below 4 °C). Samples from HD brands containing PHMG were analyzed. This study updated M_n_ and M_w_ and the degree of polymerization quantified in the last study [9]. Data on the distribution and oligomeric compound structure of the MW in PHMG were newly added here.

### 4.2. MALDI-TOF Mass Spectrometry for the Determination of PHMG Molecular Weight

Analyses of PHMG oligomers were made using matrix-assisted laser desorption/ionization–time-of-flight mass spectrometry (MALDI-TOF MS: Autoflex Speed series; Bruker Daltonics, Bremen, Germany). The average MWs were determined based on the abundance of the oligomer peaks in the MALDI-TOF MS spectra. The MALDI-TOF MS spectra were acquired in positive-ion reflectron mode. The guanidine-containing oligomers in the humidifier disinfectant samples were purified and selectively enriched using mixed-mode (strong cation exchange and reversed-phase sorbent materials) solid-phase extraction (MCX SPE, 1 mL; Waters, Milford, MA, USA) for MALDI-TOF MS analysis. The detailed purification and enrichment procedure can be found in the literature [30]. The purified and enriched PHMG oligomers were mixed with an ionic liquid matrix (ILM) to get the MALDI-TOF MS spectra. The ILM was made by dissolving 37.8 mg (0.2 mmol) of α-cyano-4-hydroxycinnamic acid (CHCA, CAS no. 28166-41-8; Sigma-Aldrich, St. Louis, MO, USA) and 15.93 μL (0.2 mmol) of 1-methyl-imidazole (CAS no. 616-47-7; Sigma-Aldrich, St. Louis, MO, USA) in methanol (a total of 1 mL solution). This ILM is known to significantly improve the spot-to-spot, shot-to-shot, and sample-to-sample reproducibility of the MALDI-TOF MS spectra. A 1 µL aliquot of the mixed ILM sample solution was deposited onto a MALDI plate. With the aid of the ILM, a homogeneous thin film of the mixed sample and ILM was formed. Nd:YAG laser light (355 nm) was irradiated at 500 Hz onto the sample spot of the MALDI plate, and a MALDI-TOF MS spectrum was acquired using 1000 laser shots. Typical operation parameters for the MALDI-TOF MS spectra acquisition were as follows: ion source 1 voltage, +19.05 kV; ion source 2 voltage, +16.70 kV; laser power percentage, 48%; pulsed ion extraction, 140 ns; lens voltage, +8.24 kV; reflector voltage, +20.99 kV; and reflector 2 voltage, +9.73 kV. The relative summed abundances of oligomer peaks appearing in the given *m/z* interval, e.g., m/z = 400~500, were denoted. In summing the abundances of oligomer peaks, the isotopic peaks, including the so-called M (mono-isotopic), M + 1 (second isotopic), and M + 2 (third isotopic) peaks, were all counted. The distribution, average, and structure of the PHMG MW and the polymerization level of PHMG were compared among the HD brands. 

## 5. Conclusions

PHMG used in HDs comprises oligomeric compounds chaining up to seven monomer units. On average, 90% of PHMG is distributed below 800 g/mol. The MW distribution, average MW, structural types of PHMG, and degree of polymerization clearly indicate that the PHMG collected from HDLI patients is an oligomeric compound. This finding verifies that the PHMG brands used in household humidifiers should not have been exempted from toxic chemical registration. Further study is necessary to examine the association between PHMG oligomeric compounds and respiratory health effects, including HDLI.

## Figures and Tables

**Figure 1 molecules-26-04490-f001:**
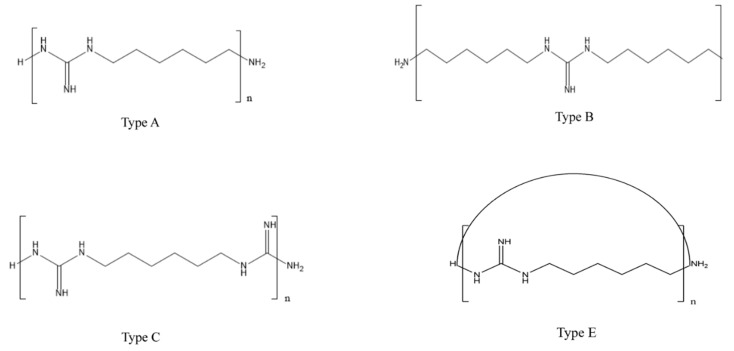
Structures of four isomeric types of PHMG oligomers in HD products, as identified in the MALDI-TOF MS spectra.

**Figure 2 molecules-26-04490-f002:**
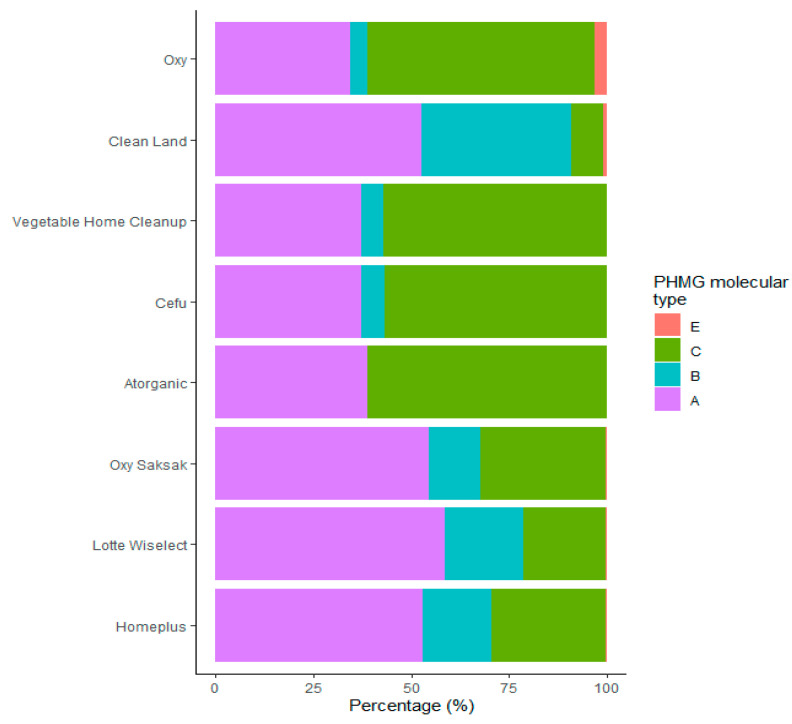
Distribution of molecular types of polyhexamethylene guanidine (PHMG) among types of humidifier disinfectant products.

**Table 1 molecules-26-04490-t001:** Number-average (M_n_) and weight-average (M_w_) molecular weights of PHMG ^2^ by brand ^1^.

Name of HD ^3^ Brand	Number of Samples ^4^	Number-Average MW (M_n_, g/mol)				Weight-Average MW (M_w,_ g/mol)				Level of Polymerization			
		Mean	SD ^5^	Min.	Max.	Mean	SD	Min.	Max.	Mean	SD	Min.	Max.
Oxy	4	514.2	32.9	465.0	532.7	534.3	22.2	501.0	547.0	3.4	0.8	2.3	3.8
Clean Land	1	422.0	NA ^6^	422.0	422.0	441.0	NA	441.0	441.0	2.1	NA	2.1	2.1
Vegetable Home Cleanup	8	523.3	30.5	483.0	587	540.0	36.4	483.0	612..0	3.1	0.9	1.0	3.7
Cefu	3	501.3	25.4	472.0	517.7	521.7	11.8	508.0	529.0	3.2	0.8	2.3	3.7
Atorganic	2	475.0	101.8	403.0	546.9	493.0	97.6	424.0	562.0	3.9	1.4	2.0	3.9
Oxy Saksak	102	552.0	65.8	435.0	692.2	569.8	61.4	460.0	714.7	2.9	1.2	1.0	4.0
Lotte Wiselect	9	547.8	63.5	475.0	673.4	566.1	54.3	511.0	682.7	2.8	1.2	1.0	3.9
Homeplus	8	491.8	41.4	444.0	532.7	518.6	30.6	475.0	549.0	3.0	0.8	2.2	3.8
*p*-value ^7^		0.07				0.08				0.96			
Total	137	542.4	64.6	403.0	692.2	560.7	59.9	424.0	714.7	2.9	1.1	1.0	4.0

^1^ These data have been added to the results from our recent study [9]. ^2^ PHMG: polyhexamethylene guanidine. ^3^ HD products were manufactured from a raw ingredient called SKYBIO 1125, containing 25% PHMG as an antimicrobial additive. ^4^ Imbalance in the number and distribution of MW (Table 2) among HD brands. ^5^ SD: standard deviation in the unit of g/mol. ^6^ NA: not available. ^7^
*p*-Value for comparing mean values from an ANOVA test, with Clean Land and Atorganic being excluded due to small sample sizes.

**Table 2 molecules-26-04490-t002:** Distribution of molecular weight in PHMG ^1^.

Name of HD ^2^ Brand	Number of Samples ^3^	Molecular Weight (MW, g/mol), %							
		400–500	500–600	600–700	700–800	800–900	900–1000	>1000	Total
SKYBIO 1125 ^4^	2	7.36	4.57	18.64	37.74	24.89	6.01	0.82	100
Oxy	7	18.40	8.92	25.40	34.65	6.78	5.36	0.50	100
Clean Land	1	53.32	29.05	11.02	6.62	0.00	0.00	0.00	100
Vegetable Home Cleanup	10	33.41	16.90	26.70	19.03	1.24	2.65	0.07	100
Cefu	6	24.09	11.50	26.01	32.32	2.20	3.63	0.24	100
Atorganic	4	52.46	7.80	21.47	18.27	0.00	0.00	0.00	100
Oxy Saksak	90	33.93	18.42	14.50	22.73	7.54	1.95	0.93	100
Lotte Wiselect	12	22.06	19.16	14.43	29.64	13.54	0.85	0.33	100
Homeplus	9	24.76	16.89	15.50	25.62	9.78	4.61	2.84	100
Total (average)	141	29.98	14.80	19.30	25.18	7.33	2.78	0.64	100

^1^ PHMG: polyhexamethylene guanidine. ^2^ HD: humidifier disinfectant. ^3^ Imbalance in number (Table 1) and distribution of MW among HD brands. ^4^ An antimicrobial product containing 25% PHMG.

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
