# Peer review of "Characteristics of the Molecular Weight of Polyhexamethylene Guanidine (PHMG) Used as a Household Humidifier Disinfectant"

_molecules, 2021, doi:10.3390/molecules26154490_

Round 1

Reviewer 1 Report

The article “Characteristics of Molecular Weight of Polyhexamethylene 2 Guanidine (PHMG) used as Household Humidifier Disinfectant, Focusing on Respiratory Health Effects” discusses the properties of PHMG found in humidifier disinfectants used in Korea. HDs were found to cause respiratory distress, resulting in HDLI and fatalities. This article is interesting and relevant to determining that HDs should have been evaluated as toxic compounds before use as HDs in commercial products. However, the article should be improved by adding statistical analyses to back up claims instead of using nonspecific language such as “far higher” and “massive discrepancy” to describe differences that need to be shown to have statistical significance in a journal article. As written, the paper only includes averages for humidifier brands and then states that there are differences without further analysis. If differences are not found to be significant using statistical testing, impact of article findings may be decreased if differences are due to normal variation within a population. Specific areas that could benefit from statistical analysis are noted in comments section below. After major revision, this article should be re-evaluated for acceptance in Molecules.

Comments:

Consider removing “Focusing on Respiratory Health Effects” from the article title, as health effects are not directly investigated in this article.

Line 26: Define “MS” at first use

Line 34: This sentence seems out of place. It’s an opinion that would be better to remove from abstract or move to the conclusions section.

Page 2, Introduction: Did lung injuries occur due to individuals misusing the HD products, or did injuries happen even when products were used as instructed? A brief sentence of how products were meant to be used could be helpful to readers to understand the problem (around line 46).

Introduction: Is the danger of PHMG due to the volatilization process? As in, is it a liquid/solid chemical that is not hazardous in its initial form, but when added to a humidifier, the chemical oligomerizes as it is volatilized and inhaled? Therefore, would part of the issue being that the humidifiers were not properly cleaned (rinsed out) to remove the product prior to use? Or does injury occur due to PHMG exposure even when not used for humidifiers? Are these chemicals safely used in other products?

Page 3: Why is SKYBIO 1125 included in Table 2 but not Table 1? Is this because Table 1 is from Ref 9? If Table 1 overlaps with Ref 9, then don’t repeat information in current paper; you can cite 9.

Table 1: Standard deviations for some brands are extremely high and outside of normal acceptance range (30%). This may be understandable for brands with only a few samples, but Oxy Saksak with 102 samples has SDs of 65. What is justification for this? Are there errors in measurement or does this show inconsistency between samples of same brand?

Table 2: Why are different numbers of samples included for each brand? Can you really get an accurate MW distribution from only 1 or 2 samples from a brand compared to 90 from another? Why are there different numbers of samples for each brand in Table 2 compared to Table 3?

Page 4, Lines 107-109: Please include statistics showing how far apart current results are from literature results for MW distributions. Is it possible these are just measurement errors? What is acceptable percent difference for these values? 

Page 5, Line 122: “Far higher” is a subjective and indefinite term. Please provide an exact value and statistical analysis justifying significance.

Page 5, Line 125: Again, please provide statistical analysis justifying the “massive discrepancy” in PHMG MWs. It is unclear how large the difference is and if it is truly significant without statistical testing.

Page 5, Line 133: How many casualties? An exact number or estimate would be helpful. How does this compare to other brands?

Page 5, Line 137: What are CMIT and MIT and how may the addition of these compounds affect toxicity of the HDs? This may be better to explain in the following paragraph.

Page 5, Line 141-142: Please be more specific about what these “differences” entail and provide ranges and examples: “different chain lengths and different isomeric molecular structures with different end groups” is very vague.

Page 5, Line 154: How similar were the MWs; how different were concentrations? Percent difference, Statistical analysis? Need to back up these claims.

Page 6, line 200: Include temperature of refrigeration.

Page 6, line 201: Please describe further how values were “updated”. Were they incorrect in previous study? Were more samples added? Was a different analysis performed? This is unclear.

Reviewer 2 Report

Comments:Household humidifier disinfectants (HD) containing Polyhexamethylene guanidine (PHMG) have caused a large number of lung injuries, including pulmonary fibrosis, in South Korea. Although there have been sufficient studies on the inhalation toxicity of PHMG, few studies have focused on whether the physical and chemical properties of PHMG in different humidifier disinfectant brands and their effects on health are different. This manuscript focuses on characterizing the MS distribution, the number-average molecular weight (Mn), the weight-average molecular weight (Mw), and the structural types of PHMG used in HD products. This study may provide new evidence for the study of respiratory toxicity of PHMG and promote the establishment of use standards. However, there still have some limitations in this research which should be addressed and discussed.

Q1. To display the distribution of molecular weight and molecular type in PHMG, it would be more intuitive to use a bar plot.

Q2. In previous studies, PHMG has been confirmed to be a polymer family containing guanidine subunits, and it has been mentioned in previous articles that there are seven subtypes and different salt types[1,2]. However, the results of this article only show the distribution of four subtypes. The reason why the distribution of the other subtypes was not mentioned should be explained in the background or discussion section of the article.

1.Park, D.U., et al., Properties of Polyhexamethylene Guanidine (PHMG) Associated with Fatal Lung Injury in Korea. Molecules, 2020. 25(14).

  1. Zhu X , Kong X , Ma S , et al. TGFβ/Smad mediated the polyhexamethyleneguanide areosol-induced irreversible pulmonary fibrosis in subchronic inhalation exposure[J]. Inhalation Toxicology, 2020, 32(11-12).

Q3. Although the author mentioned in the method section that the number of samples of different brands is extremely uneven, there are also differences in the sample size in different measurement indicators. If it is due to the method of measurement or the quality of the sample, it should be supplemented in the discussion.

Q4. In the discussion part of the article, it is mentioned that the MW of PHMG may be associated with health risk. It is pointed out that the proportion of low MW PHMG in various HD exceeds the standards established by various countries. However, the authors did not effectively discuss the difference between high MW and low MW PHMG in causing lung injury. The author had better make a supplementary explanation to this part, or enumerate the effect of the difference of molecular weight of other similar substances on its toxicity.

Q5. There are some problems in the article, such as missing punctuation, misuse, complex word order, lack of conciseness, etc. Line 111: “SKYBIO 1125 had an Mn of 1,274 g/mol, but no information on the MW distribution was provided” instead of “SKYBIO 1125 had an Mn of 1,274 g/mol but no information on the MW distribution was provided”; Line 115: “PHMG has molecular weights lower than 1000 g/mol, and the products have hexamethylene diamine (HMDA) contents lower than 1% (w/w)” instead of “PHMG has molecular weights lower than 1000 g/mol and the products have hexamethylene diamine (HMDA) contents lower than 1% (w/w)”; line 138: “In the PHMG-containing HD products, several components may be potentially related to health” instead of “In the PHMG-containing HD products are several components that may be potentially related to health”; line 234, the full stop missing; etc. A language edit is needed to improve the English grammar and syntax.

Round 2

Reviewer 1 Report

The authors have responded to all comments from the first review. The revisions to manuscript have greatly improved the clarity and scientific soundness of the article. The additional information regarding PHMG that was added to the text will make it easier for readers not familiar with the product to understand how it was used and why this led to detrimental health effects and casualties. The addition of more statistical analyses also improved the scientific soundness of the study. I recommend that the article be accepted in present form.

Author Response

Response to Reviewer 1 Comments

Comments:

The authors have responded to all comments from the first review. The revisions to manuscript have greatly improved the clarity and scientific soundness of the article. The additional information regarding PHMG that was added to the text will make it easier for readers not familiar with the product to understand how it was used and why this led to detrimental health effects and casualties. The addition of more statistical analyses also improved the scientific soundness of the study. I recommend that the article be accepted in present form.

Response: All the authors are grateful for your comments and suggestions. You have contributed considerably to improving the quality of this manuscript.

Reviewer 2 Report

This paper discusses the use of different subtypes of PHMG in different brands of humidifiers. However, PHMG also has different salt types, which might affect its toxicity. The author had better mention the distribution and toxicity of different salt types of PHMG in different brands in the background or discussion section. This will allow the article to explore PHMG in more depth and will be of great help in understanding such substances as PHMG. The following are several studies on the toxicity of different salt types of PHMG, which the author can use for reference.

[1] Zhu, Xiaoxiao et al. “TGFβ/Smad mediated the polyhexamethyleneguanide areosol-induced irreversible pulmonary fibrosis in subchronic inhalation exposure.” Inhalation toxicology vol. 32,11-12 (2020): 419-430. doi:10.1080/08958378.2020.1836091  (PHMG-hydrochloride salt)

[2] Jeong, Mi Ho et al. “Polyhexamethylene guanidine phosphate-induced upregulation of MUC5AC via activation of the TLR-p38 MAPK and JNK axis.” Chemico-biological interactions vol. 305 (2019): 119-126. doi:10.1016/j.cbi.2019.03.030 (PHMG-phosphate salt)

[3] Kim, Ha Ryong et al. “Polyhexamethylene guanidine phosphate aerosol particles induce pulmonary inflammatory and fibrotic responses.” Archives of toxicology vol. 90,3 (2016): 617-32. doi:10.1007/s00204-015-1486-9 (PHMG-phosphate salt)

Author Response

Response to Reviewer 2 Comments

Comments:

This paper discusses the use of different subtypes of PHMG in different brands of humidifiers. However, PHMG also has different salt types, which might affect its toxicity. The author had better mention the distribution and toxicity of different salt types of PHMG in different brands in the background or discussion section. This will allow the article to explore PHMG in more depth and will be of great help in understanding such substances as PHMG. The following are several studies on the toxicity of different salt types of PHMG, which the author can use for reference.

[1] Zhu, Xiaoxiao et al. “TGFβ/Smad mediated the polyhexamethyleneguanide areosol-induced irreversible pulmonary fibrosis in subchronic inhalation exposure.” Inhalation toxicology vol. 32,11-12 (2020): 419-430. doi:10.1080/08958378.2020.1836091  (PHMG-hydrochloride salt)

[2] Jeong, Mi Ho et al. “Polyhexamethylene guanidine phosphate-induced upregulation of MUC5AC via activation of the TLR-p38 MAPK and JNK axis.” Chemico-biological interactions vol. 305 (2019): 119-126. doi:10.1016/j.cbi.2019.03.030 (PHMG-phosphate salt)

[3] Kim, Ha Ryong et al. “Polyhexamethylene guanidine phosphate aerosol particles induce pulmonary inflammatory and fibrotic responses.” Archives of toxicology vol. 90,3 (2016): 617-32. doi:10.1007/s00204-015-1486-9 (PHMG-phosphate salt)

Response: All the authors are grateful for your 1st comments and suggestions. You have contributed considerably to improving the quality of this manuscript. We agree with your 2nd comments.

There are two salt types of PHMG used as HD in Korea: phosphate and hydrochloride. According to MSDS provided by company, only Vegetable Home clean up used PHMG hydrochloride. All HDLI patients responded to use HD brands containing PHMG phosphate. As far as we reviewed, no study has evaluated the difference in toxicity and MS distribution between two salt types. We think that the discussion regarding difference in toxicity between salt types is beyond this study. Instead, the necessity of further study was included with three references you recommended as follows Please line 218-219.

Further research is necessary to examine the health effects of oligomeric PHMG and the relationship between MW and the structural type and two salt types (phosphate and hydrochloride)[1-3], and toxicity of PHMG.

Reference.

  1. Zhu, X.; Kong, X.; Ma, S.; Liu, R.; Li, X.; Gao, S.; Ren, D.; Zheng, Y.; Tang, J. TGFβ/Smad mediated the polyhexamethyleneguanide areosol-induced irreversible pulmonary fibrosis in subchronic inhalation exposure. Inhalation Toxicology 2020, 32, 419-430.
  2. Jeong, M.H.; Park, Y.J.; Kim, H.R.; Chung, K.H. Polyhexamethylene guanidine phosphate-induced upregulation of MUC5AC via activation of the TLR-p38 MAPK and JNK axis. Chemico-biological interactions 2019, 305, 119-126.
  3. Kim, H.R.; Lee, K.; Park, C.W.; Song, J.A.; Park, Y.J.; Chung, K.H. Polyhexamethylene guanidine phosphate aerosol particles induce pulmonary inflammatory and fibrotic responses. Archives of toxicology 2016, 90, 617-632.